# Women Trading Sex in a U.S.-Mexico Border City: A Qualitative Study of the Barriers and Facilitators to Finding Community and Voice

**Claudia Gonzalez** [1]**, Kimberly C. Brouwer** [2,3]**, Elizabeth Reed** [4,5]**, Melanie J. Nicholls** [1,3]**, Jessica Kim** [6]**, Patricia E. Gonzalez-Zuniga** [7]**, Andrés Gaeta-Rivera** [8,9] **and Lianne A. Urada** [1,3,5,]*

[1] College of Health and Human Services, San Diego State University School of Social Work, San Diego, CA 92182, USA; Claudia.gonzalez1016@gmail.com (C.G.); melaniejnicholls@yahoo.com (M.J.N.)

[2] Department of Family Medicine and Public Health, University of California, La Jolla, CA 92093, USA; kbrouwer@ucsd.edu

[3] Department of Medicine, Division of Infectious Diseases and Global Public Health, University of California, La Jolla, CA 92093, USA

[4] Division of Health Promotion and Behavioral Science, San Diego State University School of Public Health, San Diego, CA 92182, USA; ereed@sdsu.edu

[5] Center on Gender Equity and Health, University of California, La Jolla, CA 92093, USA

[6] Center for Justice and Reconciliation, Point Loma Nazarene University, San Diego, CA 92106, USA; jkim1@pointloma.edu

[7] Casa del Centro and the Wound Clinic, 22000 Tijuana, Baja California, Mexico; lacasadelcentro1@gmail.com

[8] Instituto Chihuahuense de Salud Mental, 31000 Chihuahua, Mexico; andres.gaeta@gmail.com

[9] School of Medicine and Psychology, Universidad Autónoma de Baja California, 21289 Tijuana, Baja California, Mexico

[*] Correspondence: lurada@sdsu.edu

**Abstract:** Poverty and income inequality can increase a woman's decision to engage in risky transactional sex, and may lead to unimaginable harms, such as violence, substance use, and human trafficking. This study examines the facilitators and barriers to finding community and voice among women trading sex in Tijuana, Mexico, and what factors, such as socio-structural support, violence, and substance use, may impact their potential to engage with others, including human service providers. Sixty qualitative in-depth interviews were conducted with women trading sex in Tijuana, Mexico. Researchers met with participants for in-depth-face-to-face structured interviews. Data were coded using ATLAS.ti. Participants were aged 19–73 (mean: 37), 98% were of Mexican nationality, 90% reported trading sex independent of the control of others, with 58% identified as independent and street-based. Thirty percent of women trading sex reported substance use (excluding marijuana) and 20% reported injection drug use within 30 days. The majority reported no involvement in mobilization activities, but 85% expressed interest. However, barriers included stigma, cultural gender norms, partner violence, and privacy in regards to disclosure of sex trade involvement, moral conflict (revealing one's involvement in sex trade), involvement in substance use, human trafficking, and feeling powerless. Facilitators were having a safe space to meet, peer support, self-esteem, feeling heard, knowledge of rights, economic need to support families, and staying healthy. Findings imply the potential to go beyond mobilizing limited groups of women in the sex trade and instead involve whole community mobilization; that is, to reach and include the more vulnerable women (substance use, trafficked) in supportive services (social services, exit strategies, better healthcare opportunities, and/or education for healthcare providers to help break societal stigmas regarding women in the sex trade) and to change the status of women in society in general.

**Keywords:** women; sex trade; violence; substance use; community mobilization; human trafficking; HIV; STI

## 1. Introduction

Poverty and income inequalities increase the vulnerabilities in women, which can increase their engagement in risky transactional sex and may lead to unimaginable harms, such as substance use and human trafficking [1–4]. At the Mexico-U.S. border, financial need is a major motivation for women to initiate and continue trading sex [5]. Women are overrepresented in Mexico's informal economy due to the persistent financial crises and forces of globalization [6]. In Mexico, less than half of women participate in the formal economy, with the remaining workforce forced to engage in alternative income-generating strategies within the market economy [7]. Females trading sex in Tijuana, Mexico have been found to have low-educational attainment, low literacy, multiple financial dependents, inconsistent knowledge of HIV and sexually transmitted infections (STIs), and to have traveled from distant locations (e.g., Rural Mexico, Central and South America) in hopes of getting to the United States [4,7]. Most of these migrating women find themselves with few options for employment and financial support with limited job opportunities [5,8]. Some women in Tijuana are drawn into the sex trade because it provides a better living wage to meet their basic needs than other unskilled labor. In addition, some have flexible work schedules to care for their family [5,8].

Tijuana is the most populous city in Baja California, with a population of 1.6 million [9]. The San Diego-Tijuana border is home to one of the busiest land border crossings in the world, with over 77 million people crossings a year between both countries [10]. Tijuana continuously receives thousands of people from the U.S and from the interior of Mexico. Tijuana's transient population has supported sex and drug tourism [4,11]. In addition, thousands of families live, work, and socialize in the Mexico-U.S. border region, many with social, sexual, and family ties in both countries.

Stemming from these social problems, the interplay between poverty, migration, drug trafficking, and the threat from sexually transmitted infections (STIs) and HIV infection intersect. In Tijuana, women engaging in survival sex face an array of social problems that impede them from finding community. Research findings have indicated that two-thirds of women along the Mexico-U.S. border may be confronted by violent encounters with clients [6]. Drug use is also common within this population, and those who inject drugs have a high prevalence of STIs [12–15]. Understanding the nature of community mobilization to include those who are substance using is critical in delivering interventions for women engaging in survival sex in the border region.

This paper uses Blankenship, West, and Kershaw's [16] Community Mobilization (CM) model as an organizing framework to explain the barriers and facilitators of CM among a sample of women trading sex in Tijuana, Mexico. Community mobilization can be defined as moving beyond individual-level behavioral interventions to affect social change via social action when a group has collective identity (having a strong sense of unity with others), collective efficacy (the belief that they can work together for change), and collective agency (taking action with others on behalf of a group). Blankenship et al. [16] developed a structural intervention framework in Andhra Pradesh, India to analyze the associations between power, condom use among women in the sex trade, and how exposure to community mobilization interventions affect these associations. Power was identified through three domains: collective identity, collective efficacy, and collective agency. Blanchard et al. [17] also defined power as power within, power over, and power with. Power was measured through the individual (collective identity/power within), the group (collective efficacy/power with), and the community (collective agency/power over). Others [18–20] have integrated community mobilization (e.g., through peer leaders) into activities for women in the sex trade in Rio de Janeiro, Brazil, Manila, the Philippines, and Tanzania, Africa. Community mobilization has been found to increase engagement in clinical services, increase condom use, and decrease new HIV and gonorrhea infections in women in the sex trade [20–22].

However, little data exist on the potential for mobilization to occur among women engaging in survival sex and substance use in Mexico. One organization in Mexico City used a community mobilization model to organize women in the sex trade to protect their human rights [23], and another in Tijuana has marched for their rights against police abuse [24]. Limited studies, however, have focused

on the barriers and facilitators women in the sex trade face in finding community [23], especially those who have been excluded due to their substance use. Also, in the context of Tijuana, little research has advocated for the need to go beyond mobilizing limited groups of women in the sex trade and to instead include women in supportive services more (social services, exit strategies, better healthcare opportunities, and/or education for healthcare providers to help break societal stigmas regarding women in the sex trade) and to change the treatment of women in society in general so that they would not have to exchange sex for survival.

*1.1. Collective Identity Facilitators and Barriers*

Providing a safe and private space to meet allows people to form a *collective identity*. Women in the sex trade in India were more likely to engage in collective spaces if they had a strong sense of collective identity [25]. In two qualitative studies of women in Tijuana, those in the sex trade were willing to participate in the studies if the location was safe and convenient [5,6,26]. Having access to a private space, close to the red-light district, enabled them to meet and maintain a private identity. The common theme throughout the literature review was the need for women in the sex trade to maintain their sex trade role separate from their home life due to societal stigma and their internal moral conflict [5,8].

In Mexico, marianismo and machismo are two cultural concepts used to show the prescribed differences in power and status between men and women. The cultural concept of *machismo* can be described as male power and aggression towards women and marianismo as women's need to put everyone's desires above their own [5,6,8]. Women trading sex described a societal structure based on male dominance in the workplace and male rights to women perceived as stepping out of their traditional roles, whether by remaining unattached to a male protector or by attempting to enter the realm of unpaid labor [8].

*1.2. Collective Efficacy and Collective Agency Facilitators and Barriers*

Collective efficacy among women trading sex is the belief that they can work together for change. Collective agency is demonstrated by two or more people acting together [27]. Collective agency is best characterized by creating confidence and empowering women in the sex trade to address the problems they are facing [22,25].

Women tend to work together when their financial means are threatened or when they become aware of their human rights [23,24,28]. In Tijuana, women tend to be the sole supporters of extended families, including their own children, their parents, and the families of siblings, thus, threatening their financial means is perceived as also threatening their family's income [5,6,8].

In Mexico City, the Brigada Callejera con Apoyo a la Mujer (BCAM) (Street Brigade Supporting Women), a women-led organization, has lobbied, trained and promoted their rights to combat discrimination, repression from police, and removal of resources for women in the sex trade since 1995 [23]. In Tijuana, a group of women in the sex trade organized and marched to the *Palacio* Municipal (Government Building) to demand that their rights be heard after the Callejon (alley) was shut down [24]. In May 2016, this same group of women organized to file a complaint in which they accused a female municipal police officer of extortion [28]. These groups have mobilized because physical, verbal, and psychological violence from clients and police can contribute to women's vulnerability [6,25,29,30]. Violence is sometimes normalized among women in the sex trade; they may think it comes along with shame and stigmatization [6,29]. In addition, women trading sex face discrimination and social exclusion from professions that are supposed to treat everyone fairly. For example, women trading sex have felt a distinct repressive behavior towards them from doctors, nurses, and police officers [30,31]. The drive towards collective efficacy and collective agency have overlapping tendencies in which the women feel a collective urge to come together to create change amongst themselves and their peers. However, substance-using women are often excluded from such groups, and women who are trafficked, e.g., by cartels, are not usually a part of any other organization.

Although mobilization of women in the sex trade in Mexico occurs, the extent and nature of mobilization among the most vulnerable women (e.g., substance-using) in Tijuana is not well understood. The objectives of the current study are to explore the facilitators and barriers to finding community and voice among women trading sex in Tijuana, Mexico in order to inform the social supports (e.g., services, exit strategies) needed for the most vulnerable populations of women in the sex trade (e.g., trafficked, substance using), many of whom are typically excluded by others or hidden from reach.

## 2. Methods

### 2.1. Ethical Review Board Approvals

Ethical approval for this study was granted by the Institutional Review Boards (IRB#130498) of the University of California, San Diego, San Diego State University, and of Mexico's Colegio de la Frontera Norte.

### 2.2. Study Sample and Data Collection Procedures

This qualitative study analyzes data from Mujeres Unidas: Community Mobilization and HIV/STI Epidemiology of Females in the sex trade in Mexico, funded by the National Institute on Drug Abuse (NIDA) (K01DA036439). The key informant interviews (*n* = 60) were drawn from participants in the Mapa de Salud, NIDA project (R01DA028692), a longitudinal study investigating social, structural, and spatial determinants of HIV/STI and substance use risk among women trading sex in Tijuana and Ciudad Juarez, Mexico. However, the qualitative study only involved the women in Tijuana. In the parent study, Mapa de Salud, women in the sex trade were stratified by work location (e.g., bars, brothels, hotels, alleys, and street corners) and type of drugs used, to get a systematic sample of the study areas and an equal number of participants from street and venue settings. To be eligible for the parent study, the participants had to be at least 18 years old; had exchanged sex for money, goods, or shelter four or more times with four or more clients in the previous 30 days; be willing to undergo behavioral surveys (CAPI: computer-assisted personal interviewing survey system) and HIV/STI tests in the parent grant; agree to receive STI treatment if positive from the medical test; and lived in Tijuana, Mexico with no plans to leave the area within 18 months of the first interview date. Those not using substances were included, as well as those using drugs, because the non-substance-using women were compared with those who used drugs in both the parent grant's survey and the qualitative in-depth survey data collection. Participants received an equivalent to USD$20 reimbursement for their time and transportation to the study site, condoms, referrals to substance use and HIV treatments, and counseling and literature on how to clean needles/syringes, as appropriate.

Researchers and trained bilingual staff skilled in HIV, the sex trade, and ethical study protocols met with participants in a centrally located study office near to where the women lived and worked for in-depth-face-to-face structured 60–90 min interviews (2013–2014). Outreach staff from the community initially brought them in or told them the location of the office. During the qualitative interviews, the researcher used open-ended questions, e.g., to understand the participants' sense of collective identity, collective efficacy, and collective agency. For the interview guide, see Appendix A. The parent grant provided socio-demographics to contextualize these interviews. This paper presents the analysis of the qualitative data derived from these 60 in-depth-face-to-face interviews.

### 2.3. Data Analysis

The interviews were audio-taped and transcribed verbatim by professional transcriptionist and translated by a certified translator into English (without identifiers). Bilingual staff revised translations. The lead investigators reviewed the data to develop an inductive and iterative list of mutually exclusive, but possibly linked, codes. Lead researcher and trained bilingual research assistant staff worked independently to code text files for data analysis related to the community mobilization measures.

Inter-coder reliability across coding was reached based on the standard approach from Carey, Morgan, and Oxtoby [32]. ATLAS.ti, a qualitative data analysis software package, was used to code and manage the data [33]. Data were compared between and across all participants.

Community mobilization, barriers to mobilization, and power and agency were the three main domains used to code the English translated interviews to assess collective identity, collective efficacy, and collective agency, based on Blankenship, West, and Kershaw's [8] Community Mobilization model (2008). Community mobilization touched on social support, support from agencies, motivations to mobilize, and women's needs, interests, and availability in programs. Barriers to mobilization included areas such as isolation and the lack of trust, rivalries among women in the sex trade, such as those using substances and those who are not, and secrecy. Power and agency covered women in the sex trade and their sense of autonomy, self-esteem or self-efficacy, powerlessness and lack of control, how they manage their identities, and their knowledge of human rights. For a full summary of the domains and codes used in this analysis, see Table 1.

**Table 1.** Community mobilization domains/codes for data analysis, women in the sex trade in Tijuana, Mexico.

| **Collective Identity Facilitators** | **Definitions** |
| --- | --- |
| Safe and Confidential Space for Finding Community | Renting or delineating territories on streets or other spaces used |
| Agency Support/Services | Having support from an agency that can also provide services |
| Social Support for Self-Esteem/Efficacy | Although these are distinct concepts, we group them together because they can be difficult to disentangle. Here, we want to capture examples of women's sense of self-worth (or lack thereof) and/or women's belief in their own ability to succeed or achieve goals. |
| **Collective Identity Barriers** | **Definitions** |
| Cultural Expectations of Gender Roles in Mexico | Covers how straying from traditional roles can make it difficult to mobilize and access resources |
| Substance Use Stigma | Reports of women's isolation from other women in the sex trade or from services such that it serves as a barrier to mobilization. |
| **Collective Efficacy Facilitators** | **Definitions** |
| Knowledge of Human Rights | Covers basic knowledge, or lack of knowledge, of broader human rights or women's rights. May be a barrier to or motivation for mobilization |
| When Economic Means are Threatened | Includes any type of support from peers, agencies, or social services to help with economic means. |
| **Collective Efficacy Barriers** | **Definitions** |
| Human Trafficking | Defined as concern about victims of trafficking as they experienced powerlessness and loss of control, which made it more difficult for them to access resources to get help. |
| **Collective Agency Facilitators** | **Definitions** |
| Staying Healthy | Accessing resources or mobilizing with others to protect health |
| Feeling Heard | Having support from peers and agencies where they feel listened to |
| **Collective Agency Barriers** | **Definitions** |
| Physical/verbal/psychological violence from clients, the police, and intimate partners [pimp] | Describes instances where women have/have not had support from other women in the sex trade to deal with problem clients or violence. Social support could take many forms: emotional, financial, etc. |
| Societal Stigma | Reports of women's isolation from services and their personal relationships such that it serves as a barrier to mobilization. |

## 3. Results

Drawing from the parent study's baseline survey data, most of the participants in this qualitative study were of Mexican nationality (98%) and lived in Tijuana; their ages ranged from 19–73 years, with a mean of 37. The majority of the participants had less than a high school education (88%) and more than half (63%) were either never married, divorced/separated, or widowed. Most of the participants reported having children (93%). Just over half of those qualitatively interviewed identified as independent and street-based in the sex trade (58%) at the time of the interview. The remaining participants stated working as escorts, bar companion, or exotic dancers. Most of the participants (90%) reported trading sex independently (not having a pimp), with the remaining indicating that they had to pay a percentage to a pimp, manager, or administrator. Although STI clinic attendance and sex trade registration and STI testing in Tijuana is mandatory for those trading sex, most participants stated not being registered or else they had an expired permit with Tijuana's health department. The sample included 30% who reported substance use (excluding marijuana) and 20% persons who injected drugs in the past 30 days. Additionally, 42% had STIs (gonorrhea, chlamydia, syphilis, and/or HIV); 52% of the 60 interviewed said they had been physically abused by someone; and 17% reported being physically abused by a client at some point. These were slightly higher than the overall baseline percentages (49% and 16%, respectively).

Most of the women had never participated in mobilization activities. However, when probed during qualitative interviews about their interest in finding community and voice, 85% of the participants expressed interest. The following themes emerged related to the potential for women to create community based on their collective identity, collective efficacy, and collective agency (Blankenship et al. 2008, [16]).

### 3.1. Collective Identity

Only 15% of the participants said they had participated in mobilization activities in the past; however, the majority had not attended a meeting with others who traded sex. Therefore, most expressed the following potential facilitators towards collective identity, identifying with others who trade sex, through: space for creating community, peer and professional support for client problems and violence, and social support for self-esteem. Barriers were cultural expectations of gender roles in Mexico and substance use.

#### 3.1.1. Potential Facilitators for Collective Identity

Safe and confidential space for finding community. Most of the women feared having their sex trade activity revealed because they had not disclosed it to family members. Having a space to meet close to the place where they lived and worked (i.e., red-light district) enabled them to keep their identities private and encouraged them to voice their concerns. A 41 years old woman trading sex on the street said, "[If] I have a room, ... and see that nothing got out [i.e., privacy]".

Women expressed how having a space to create a community among them would help them meet with their peers to share concerns about client problems and violence and to find ways to help each other. Having a space to meet near their places of work also helped them build a strong sense of cohesion and trust among themselves.

> "We would tell each other, and we would support each other ... we would advise each other on difficult situations we go through or what we are going through ... " [Age 33, fichera/charges for drink client buys]

> "Tell our stories, gather, and tell our problems ... all the problems we go through ... help each other ... to see what we could solve or come to some agreement and be able to be better". [Age 26, call girl/escort]

Lack of professional intervention when encountering physical/psychological violence from clients was expressed. Many felt it was important to have immediate access to professional resources in case they experienced physical/psychological violence from clients.

"I have several acquaintances that really need to talk, more than anything that there be advice, in case something happened, we know what to do . . . besides the diseases, if they get hurt physically, whom to go to . . . some advice or something" [Age 35, trading sex on the street]

"How to defend themselves . . . To talk to someone...for example, a situation happened to me, but I didn't know where to go or who to get close to . . . but yes, the reality of knowing where to go and who to go with that can help them". [Age 21, did not disclose work location]

They also shared the importance of having information related to STIs for all.

"... I think we should make a group. Have conversations about diseases and all of that. Why? Because the girls sometimes don't go get the card and they might be transmitting [STIs] to other people . . . The ones that are sick . . . because they don't get the card, they are not afraid . . . maybe an orientation [is needed] [Age 21, did not disclose work location]

"Well [to learn] how to take care of ourselves, from diseases and all of that. They can take information . . . about infections that can be transmitted, not just AIDS. Other diseases . . . that are also severe . . . or not that severe . . . how you can prevent them and if you can prevent them . . . Have female doctors . . . that are specialized . . . [teach them]" [Age 22, call girl/escort]

Social support for self-esteem. Telling their stories and feeling heard allowed the women trading sex to increase their confidence and self-respect. They expressed interest in having a support group where they could feel validated as "human beings". Maintaining their role separate from their home life obliges them to isolate from supportive systems (e.g., family, friends, social services).

"Like more than anything, they need to be able to stand-up, their self-esteem. And to say that they are worth as a person and like a human being . . . that they don't have to be there being lessened or allowing someone to treat them bad . . . because to have them there is in some way to be locked-up . . . it's like a prison". [Age 27, trading sex on the street]

3.1.2. Potential Barriers for Collective Identity

Cultural expectations of gender roles in Mexico. Many of the women maintain a separation between their sex trade and substance use activity and their home life due to the internal moral conflict of acting non-traditionally as a female in traditional Mexican society. Women who step out of traditional gender roles experience internal moral conflicts, which lower their self-confidence and increase vulnerability to physical and psychological abuse from men and society.

"That one as a woman matters, that regardless of what you work in you always, always have values . . . that it's not our fault for having fallen here. That many of us would want to have a different life . . . yet your values, your principals, your morals . . . your female values are by the ground . . . [Being treated poorly by customers also] keeps lowering your morals until they hit the ground. So, if there was something or someone, or a group of people that tell you, 'you are valuable' . . . " [Age 21, did not disclose work location]

" . . . A man can be an addict and a saying [referring to a cultural belief] 'because he works . . . poor him . . . to stay awake, that's why he does it', . . . the woman can't because it looks bad. Or . . . [if] he is a man, they don't look as bad as the woman drugging herself . . . " [Age 21, call girl/escort]

Substance use stigma. Women who traded sex also stratified themselves by "non-drug users, drug user/non-injecting, and persons who inject drugs (PWID)". They felt the need to separate themselves from others whom they saw as "less stable" (i.e., non-drug user vs. drug user). Those who used drugs (i.e., inhale, smoke, snort) but did not inject-drugs expressed strong negative feelings towards those who injected drugs. Those who injected drugs felt further discriminated and rejected from others and society.

> " . . . when [drug] is injected they see you even worse . . . the discrimination is too much . . . ". [Age 35, trading sex on the street and injecting drugs]

> "I don't know of . . . any organization that is . . . an organization for us. Simply because . . . they discriminate . . . give us ugly looks... People take advantage of us, belittle us . . . that's what they do every day with those of us that are addicts. Or they close the door on you, or they yell at you, or they treat you badly". [Age 48, trading sex on the street and injecting drugs]

> "They look repulsive [those that inject drugs . . . I don't have a relationship [with them] . . . because their drug is different than mine . . . ". [Age 27, trades sex, inhales drugs]

### 3.2. Collective Efficacy

What motivated women to come together to create change? The following *collective efficacy* themes emerged from the interviews as facilitators of CM: knowledge of human rights and mobilizing when their economic means were threatened. Collective efficacy barriers included: human trafficking, powerlessness, and access to resources.

### 3.2.1. Potential Facilitators for Collective Efficacy

Knowledge of human rights. The perception to take care of themselves and build community with other women developed as they became aware of their human rights. Those who knew they had rights were more likely to refer to a situation in which they stood up for themselves and demanded respect as human beings. Females in the sex trade who had previously organized during the *callejon* [alley] shutdown were more likely to be informed about their human rights.

> "To respect our rights as human beings . . . that there would be a little more support, normally we have problems with the ones that make government changes . . . they want to move us [remove them from the alley]. They [also] want to raise the fee [health card to trade sex] . . . ". [Age 29, trading sex on the street]

> "We said, why don't they put human rights office here? Because they greatly violate our rights . . . even from the same clients . . . police rob them . . . but if we get together yes . . . ". [Age 53, brothel worker]

When their economic means are threatened. Those trading sex in Tijuana tended to be the sole supporters of extended families, including their own children and parents. This responsibility creates an internal drive to act when their economic means are threatened. Several of the participants reflected upon times when those trading sex had to demand their right to be on the streets.

> "Look recently . . . they started to tell us, the police and the ones from regulations, that we couldn't stand on the sidewalk . . . a lot of us got together . . . we went to the Palacio Municipal [Government Building] . . . because it was going to affect us". [Age 46, street worker]

Due to the economic situation in the city, the women trading sex in Tijuana seemed to be trapped trading sex because they had no other option.

### 3.2.2. Potential Barriers for Collective Efficacy

Human trafficking. Participants were concerned about victims of trafficking as they experienced powerlessness and loss of control, which made it more difficult for them to access resources to get help. Some women had difficulty accessing services due to psychological trauma, particularly among those with previous experiences of human trafficking. Respondents used the terms pimp and husband interchangeably when they referred to someone influencing control over their choice about attending groups with other women. The women also referred to the "men who take care of the trata de blancas" for men they saw trafficking women.

"the majority [have pimps that] . . . watch over them . . . ask them for a fee . . . [if] 'a lot of time has gone by' . . . they will hit you...sometimes one doesn't want to work here [by choice] . . . some of us do it out of necessity, other out of duty, and other by force. They wouldn't necessarily come in . . . whatever topic related to the job . . . it would be good for you to let them feel confident and it's the start for them to talk. Then, logically the topic about the job is going to come up. But it would be good . . . to have them see that they have values . . . find someone like you to trust, because you guys have knowledge to say . . . 'well they are abusing you, they are keeping you under force in a place . . . look I can guide you, I can take you where they will help you and everything is going to be fine.' That would be like a start . . . ". [Age 21, did not disclose work location]

### 3.3. Collective Agency

Women were asked questions to describe situations in which they felt social cohesion with other women and/or felt the need to act on behalf of women selling sex to exercise their collective power. The following *collective agency* facilitator themes emerged from the interviews: to protect themselves and "stay healthy" for their family and to feel heard. Barriers for collective agency included physical/verbal/psychological violence from clients and the police, and societal stigma towards those selling sex.

### 3.3.1. Potential Facilitators for Collective Agency

Staying healthy. A motivating factor to unite with others was to protect themselves and "stay healthy" for their family. Women selling sex were likely to be the sole means of economic support for their children and to have an internal drive to "stay healthy" for their family. Many saw their families as a source of motivation to engage in healthy behaviors, such as condom use and health information dissemination.

"About AIDS . . . sometimes one gets depressed and one goes down and you say, 'ah, well I'm going to die of something,' It shouldn't be like that because there are those that depend on us and we are just thinking about ourselves . . . I remember my children and it's the only thing I have, it's the only thing that motivates me . . . ". [Age 27, selling sex on the street]

Feeling heard. Experience in previous groups, along with feeling heard one-on-one, also motivated the women to engage and feel connected with others.

"I would unwind [with] so much talking . . . we would even talk about our lives . . . like how we've been and how we want to move forward. And well . . . more than anything . . . they would support us . . . help us relax . . . It would be good that they [women selling sex] had a talk to motivate them, to be different individuals . . . they are lonely, they feel lonely . . . and they don't know what to do and that is why they go as far . . . sometimes with drugs, and sometimes drinking . . . that is because they feel like that. It's because . . . when you are down . . . you arrive to an extreme . . . you get depressed". [Age 39, selling sex on the street]

"We are discriminated . . . they [society] lower your morale and when I talk to you guys [interviewers] . . . you [help] raise our self-esteem". [Age 35, selling sex on the street]

3.3.2. Potential Barriers for Collective Agency

Physical/verbal/psychological violence from clients, the police, and intimate partners [pimp]. Women experience verbal, physical, and psychological abuse from male partners, the police, and clients, which contributes to their vulnerability and powerlessness to mobilize. Violence is normalized among those selling sex given their work environment and the absence of support from law enforcement. This contributes to women having a sense of helplessness. Intimate partners who often act as pimps [padrotes] often exerts control over the women's lives.

"The other time . . . he hit my face [client] . . . it was something very ugly . . . I don't want to remember . . . we did have [sexual] intercourse . . . besides hitting me, I felt very humiliated . . . I felt that my values as a woman . . . were in the ground". [Age 27, selling sex on the street]

Q: Do you have an idea what topics would be important or interesting so that it doesn't seem like a waste of time [talking about meetings]?

A: About being careful with people that want to attack you, abuse you . . . [I've seen some of that] have some partner, some person next to them that forces them to work in this, . . . they are being forced, sexually exploited by someone . . . [The women look] beat up or worried or they aren't comfortable. And if I talk to them they give me a comment but [they will say] 'I can't say anything else or talk to you or have enough time,' I feel impotent for not being able to help someone. I would want to but if she doesn't want to . . . what can I do?". [Age 29, selling sex on the street]

Police violence was commonly experienced by those in the sex trade. The police reportedly abused their power and asked for money as bribes.

"I'd give them [policemen] more education, more ethics . . . courses so that they learn [how] to treat women, people . . . that they aren't so corrupt, so dishonest. That they don't abuse me... But there is abuse now, everywhere. Even in the conectas [location where they sell drugs] . . . men will abuse women. [the police] would stop us all the time because you could tell that we used drugs . . . [the police would say] 'we are going to take you to the judge and we are going to tell her that you had drugs.' [I said] 'you can't do that.' [they said] it's your word against mine and I'm a cop.' [they said] . . . they wanted money . . . ". [Age 22, call girl/escort]

"I had a federal highway police . . . he basically raped me. Why? Because it's one thing to give consent to something and selling something . . . but someone that forces you to do things that you don't want to . . . no one helped me". [Age 31, dance companion]

Societal stigma. The women experienced consistent stigmatizing behaviors through everyday interactions in their home and community, which made it difficult for them to act on behalf of those who were selling sex.

"Women who trade sex are there because of certain circumstances, many are single moms, we have a lot of responsibilities . . . others have private lives and they don't want anyone to find out . . . society is very cruel and they criticize you". [Age 31, dance companion]

"Many people pass by in their cars . . . or walking . . . they stare at you, talking about us and laugh. . . . Sometimes we go to the doctor or . . . they'll say: 'do you get drugged' or one will say where one works . . . and if you tell them [where you work] . . . they will start to treat you badly". [Age 51, trading sex on the street]

## 4. Discussion

This paper examines the sense of collective identity, collective efficacy, and collective agency among a sample of women in the sex trade and has found that most had not participated in community mobilization. Although 85% expressed a desire to find community and get support from others in their situation, human trafficking and other issues often presented a barrier. Finding community and a voice, feeling heard, and getting out of isolation, shame, guilt, and darkness were themes. To be empowered, strengthened, and to become resilient enough to move forward were their desires. Both substance-using and non-substance-using women in the sex trade expressed the need for less stigma from society and for better supportive services. Substance-using and trafficked women, and even those who did not experience either of these, were not involved in organizations of women and preferred an exit strategy from their current situations, including a desire for greater assistance from government and non-government healthcare and social service providers or anyone who would listen to them without judgement. Therefore, we move from solely looking through a public health lens to shift the narrative to include a social welfare lens that takes whole community mobilization into consideration, or the need for greater assistance from the entire community to change the conditions of vulnerable women in Tijuana.

Findings indicate that having a private space to gather near to where they work is important for collective identity to occur. For women in the sex trade, keeping their sex trade involvement and home life separate is important due to societal stigma and internal moral conflicts associated with traditional gender roles in Mexico [5,6,8]. The cognitive dissonance created by these internal moral conflicts produces a constant state of powerlessness among them. Women in the sex trade expressed a strong interest in support groups where they could share their concerns and receive support about client problems, especially violence, theft, and harassment against them. The constant state of powerlessness and isolation reported by many reflected their request for peer-to-peer groups to boost their self-esteem and help them feel validated and treated like "human beings". Peer-led education programs involving women in the sex trade have been found to reduce internalized stigma and help empower them to confront judgements from others [34]. These women in Tijuana largely did not benefit from human rights movements that only benefited the members of the organizations. A large percentage of women in the sex trade are excluded or never reached due to substance use or being trafficked by cartels. This paper closes the gap in the literature that has not included these populations and advocates for greater assistance for these women.

Cultural expectations of gender roles in Mexico and moral identity are highly correlated with the disempowered identity of women in the sex trade. Many don't establish relationships with others in their situation due to the social stigma related to trading sex. An interesting finding was the segregation that existed among, not only non-substance using and substance-using women, but also between those who injected drugs and those who did not. Women felt a need to separate themselves from others whom they saw as less stable [8]. Those who injected drugs were more likely to articulate lower self-esteem and shaming from other women who traded sex who perceived them as less worthy. Other studies have shown how individuals involved in substance use were likely to bring each other down and become complicit in their ongoing subordination [35]. Previous research found that Tijuana women who traded sex and reported positive expressions of support and inclusion related to promoting recovery could improve their substance using recovery goals [36]. Future practice should emphasize social support, both within the family and outside the family, for women in the sex trade who are in recovery or trying to quit using substances.

In our study population, financial need was the major motivation for women to participate and continue selling sex due to their low educational attainment and multiple financial dependents. They found themselves with few options for employment [5,8]. When their economic means were threatened, they expressed an internal drive to mobilize (i.e., collective efficacy). Participants who were aware of their human rights were more likely to consider uniting with other women for change. Community saving groups run by women in the sex trade have been found to increase empowerment,

both economically and in the community, while also raising collective identity [19]. Conditional cash transfer interventions have also been found to empower decisions on work logistics for women in the sex trade and reduce risky sexual behavior [37]. This suggests that interventions that involve women trading sex can help reduce the threat to economic means and create more solidarity within the group.

Most of the participants had children, and "staying healthy" for their family was an important factor for mutual aid and social cohesion (i.e., collective agency) to occur. The data presented in this study reveals women's acknowledgement of the importance of condom use among themselves and their peers as a protective factor against HIV and STIs. However, the stigma associated with trading sex was a major barrier to seeking services and gathering together, given societal gender expectations. Participants experienced stigmatizing behaviors from social service and medical professionals, which could further prohibit them from accessing health and social services. However, peer support groups have been found to increase clinic visits and help empower women to stand up to health care professionals who exhibit stigmatizing behaviors [21,22,34].

One of the most consistent barriers to finding community throughout the data was the physical and psychological violence the women experienced from clients, intimate partners, and the police. Participants described male dominance in the workplace and from intimate partners who exerted control over their life. For women in the sex trade, violence was expected given their work environment, unreliable support from law enforcement, and their disempowered identity, having stepped out of their traditional gender role in Mexican society [6,38]. The physical and psychological violence experienced by these women in the sex trade contributed to their feelings of vulnerability and powerlessness to mobilize. Findings from other studies [17–20] indicate that community mobilization possesses benefits for empowering women in the sex trade, both individually and collectively. However, in the Tijuana context, we argue that greater supportive services from government and non-government organizations are necessary once the women find community and voice, especially for those who are trafficked or contend with substance use and therefore do not participate in community mobilization activities with others.

An important barrier to finding community among the women was human trafficking. Several of the participants indicated that a major barrier to organizing with other women was the presence of pimps or "husbands". During the interviews, the respondents interchangeably used the term pimp (padrote) or husband when referring to the person influencing control over their lives. The participants described those with pimps or husbands as an isolated group that needed help accessing services due to their victimization. Respondents stated that victims had no control over their lives and felt hopeless about their future. Some of the respondents worried that, because this population is isolated and without access to health services (e.g., condoms), they could increase the risk of infections to other persons. This information reflects previous research findings of commercially sexually exploited children (CSEC) who relied on third parties (e.g., pimps, bar managers) or were in isolated settings due to their age, putting them at greater risk for exploitation and with no access to health or victim resources [39,40]. In Mexico, pimping and prostitution of minors under age 18 is illegal, and classified as human trafficking; however, these offenses are widely practiced in collaboration with or with the knowledge of the police [41], often involving the Mexican cartels. This study did not interview CSEC in Tijuana, a potential area of future research on human trafficking, especially considering the visible girls standing inside and outside the night clubs and bars.

The implications of this study show that women in the sex trade have a variety of needs that have been unmet. Findings provide insight into the areas that can be addressed, such as social support, stigma reduction, and economic stability. Further studies are needed that focus on interventions for women in the sex trade who are also using substances to further provide support to this vulnerable group. Prevention and intervention strategies will need to focus on strategies that work on multiple levels to better address the needs of this population [42,43].

*Limitations*

This qualitative study is not generalizable to the larger population and/or to every woman trading sex in Tijuana, Mexico. However, our intent was to include and further understand the potential to mobilize substance-using females especially, a group commonly excluded from networks of women in the sex trade who have organized around their human rights. In addition, this study is not generalizable to all women trading sex due to cultural limitations. The traditional gender roles of Mexico, described in this paper, of male power (machismo) and female compliance (marianismo) may not be prominent in other global settings [5,6,8].

The method of translating from Spanish to English by multiple translators sometimes resulted in some concepts getting lost in the translation. However, the first author of this paper had a more nuanced understanding of the language, culture, and context of the Tijuana population (e.g., cultural norms and gender dimensions) and therefore was able to more accurately interpret results directly from the Spanish transcripts. Members of organized groups of non-substance using women in the sex trade in Mexico did not participate in this study and therefore answers may not accurately reflect the experiences of all who may be more mobilized.

## 5. Conclusions

The results from this study suggest that Tijuana women who trade sex have a clear potential to find community and a voice and motivations to do so; namely to get support, boost self-esteem and self-worth, and reduce vulnerability and harm. Through a whole community mobilization framework (i.e., collective identity, collective efficacy, collective agency involving social service supports), we have explained how women in the sex trade might overcome the isolation and the structural and social conditions that increase their vulnerability to health risks and violence. Our findings show that many women in the sex trade have not been included in other mobilization efforts partly due to the levels of stigma, even within the community of women in the sex trade. Substance-using women, women who are trafficked, and CSEC youth are some who could use more supportive services such as social services, exit strategies, better healthcare opportunities, and/or education from healthcare providers to help break societal stigmas regarding women in the sex trade. A higher aim to change the status of women in society in general is needed to break the cycles of women who feel trapped in the sex trade.

To reach the most vulnerable women in the sex trade, the findings from this paper show that peer support groups are desired by the women to allow them to meet at a location outside of their workplace and residences, at a community agency or clinic, in order to get out of isolation and their dire situations. Then, instead of just small bands of women having their voices heard, a greater multi-sectoral response is needed, such as the collective impact model [44] of the San Diego Regional Human Trafficking-CSEC Advisory Council in California, the U.S., which involves nine sectors: child welfare, community, education, health, law enforcement, prosecution, research and data, victim services, and survivor voices [45].

**Author Contributions:** Conceptualization, L.A.U., C.G., and J.K.; data curation, L.A.U., C.G., K.C.B., E.R., and A.G.-R.; formal analysis, C.G. and L.A.U.; funding acquisition, L.A.U., K.C.B., and E.R.; investigation, L.A.U., C.G., K.C.B., E.R., and A.G.-R.; methodology, L.A.U., K.C.B., E.R., and A.G.-R.; project administration, L.A.U., K.C.B., E.R., and A.G.-R.; resources, L.A.U., K.C.B., and E.R.; software, L.A.U., K.C.B., E.R., and A.G.-R.; supervision, L.A.U., K.C.B., E.R., and A.G.-R.; validation, L.A.U., C.G., A.G.-R., J.K., and P.E.G.-Z.; visualization, C.G., L.A.U., and M.J.N.; writing—original draft, C.G., L.A.U., and M.J.N.; writing—review and editing: C.G., L.A.U., M.J.N., K.C.B., E.R., A.G.-R., P.E.G.-Z., and J.K. All authors have read and agreed to the published version of the manuscript.

**Funding:** This project was funded by the National Institute on Drug Abuse (K01DA036439, Urada; R01DA028692, Brouwer), the National Institute of Mental Health (K01MH099969, Reed), the Fogarty International Center of the National Institutes of Health (NIH) under Award Numbers D43TW009343 and R25TW009343, and the University of California Global Health Institute (UCGHI).

**Acknowledgments:** We are grateful to the Mujeres Unidas and Mapa de Salud staff/data collection and analysis team: Alondra Beaz, Ricardo Vera, Marissa Salazar, Brooke West, and to the women who participated in this study.

**Conflicts of Interest:** The authors declare no conflict of interest.

**Appendix A. Interview Guide**

Collective Identity (developing a sense of community)

(1)   How do you prefer to identify yourself when trading sex? How do other people's feelings about sex trade and how they refer to those who trade sex affect you? For example, discrimination and stigma? What changes in government policy, improved human rights, or improved access to healthcare do you want to see?

Collective Efficacy (coming together if a problem affected others)

(2)   If there were a problem that affected all or most women trading sex, would they come together to address it? How about substance using women trading sex? What kinds of problems? Where and how would they or do they meet, and with whom, and specifically for substance using women?

Collective Agency (social cohesion and mutual aid)

(3)   Are you aware of sex trade collectives in Tijuana? Have women in the sex trade mobilized or met together to address the problems they face? How about substance using women trading sex? What have they accomplished? What do you think community mobilization offers? How about for substance using women in the sex trade?

(4)   Are you aware of and have you participated in any group, workshop, training/class, activity, rally, or other meeting where other women in the sex trade were present? How about substance using women in the sex trade? Please describe the activities and conditions around your participation. If you did not participate, would you be interested in participating? What stops you or other women in the sex trade from participating? How about for substance using women in the sex trade? What has helped or would help you participate?

(5)   Can you describe the isolation you may currently feel, and how you may want to reduce it?

(6)   What are the health risks women in the sex trade face that might be helped by meeting with others?

    (a)   Policy environment: workplace practices/policies
    (b)   Physical environment: border migration issues
    (c)   Social environment: interaction with peers, managers, pimps, sexual partners, clients

Alcohol and Drug use

(7)   What do you think substance using women trading sex need in terms of advocacy? What concerns or issues do substance using women trading sex have?

(8)   Do you think drugs or alcohol affect a women's ability to participate in mobilization or to meet with other women trading sex about the problems or issues they face? Could community mobilization help motivate substance users to recover from drug use?

(9)   What unique strategies are required for substance using women trading sex to mobilize?

(10)   Are you aware of any sex work organization that has included substance using women? Does the type of drug or alcohol use affect one's ability to participate?

HIV/STIs

(11)   Has mobilizing/meeting with other women trading sex helped lower the risk of HIV/STIs among them? Among substance using women trading sex?

(12)   What are the necessary conditions in order for women trading sex to mobilize around their health and rights? For substance using women trading sex to mobilize? What next steps do you recommend to enable women trading sex and substance using women trading sex to mobilize?

(13) Why and how have women trading sex in Tijuana mobilized for improved human rights and access to healthcare, changes in government policy, opportunities to make more income (e.g., via group loans), and reduced isolation and abuse/violence? What is the potential and what are the barriers to mobilize around these issues for substance using women trading sex?

Policing

(14) What problems do you face from the police? [Probes: arrest, threat of arrest, put in lock-up/jail, physical and sexual violence or verbal abuse].

(15) Is violence/abuse from police a common problem for you or threat to you? Can you tell me about an incident when a police officer has beaten you (e.g., hit, slapped, pushed, kicked, punched, choked, or burned you)? Or threatened to hurt you? Or made you fear for your safety in some other way? What did you do? Do you seek other help when faced with police violence/abuse? What strategies do you use in general to avoid police violence? Are there ways you think working with a group of others could help you in preventing this violence? i.e., Can women join together to support each other, e.g., to watch children when arrested, provide bail money etc. Other ways?

(16) Did you ever help a peer when she faced a police problem?

Experiences of violence and strategies used to prevent violence

(17) Is violence/abuse from clients a common problem for you or threat to you?

(18) Can you tell me about an incident that is most memorable when a client forced you to have sex when you did not want to? Or forced you to engage sexually with him in a way that you did not want? What happened?

(19) Are there places where you feel most at risk for violence from clients? (probe: club, street, hotel, places where you are alone with the client; places where there is a lot of substance use occurring or certain types of drug use) Are there certain types of clients that are more risky in terms of being violent? (probe: drug use, clients from Mexico versus elsewhere)?

(20) Do you seek others' help (peers) when faced with client violence/abuse? Are there strategies you use in general to try to avoid client violence? If you were part of a larger group of peers whose goal was to support each other, are there ways you could work together to help protect each other from violence perpetrated by clients?

(21) What about violence from others? Have you experienced violence from others? (male partners, club managers, others who work in the club, men on the street, neighbors, etc.) Can you tell me about an incident when someone has beaten you (e.g., hit, slapped, pushed, kicked, punched, choked, or burned you)? Or threatened to hurt you? Or made you fear for your safety in some other way? What did you do? Do you seek help from peers when faced with violence from rowdies/others? What strategies do you use in general to avoid violence? How did the violence affect you? Have you ever had an injury because of a violent incident? (describe)

(22) Are there services available for women who face violence? [Probe for violence related to sex trade, violence from husbands or partners, or violence from police]? What could help address issues of violence?

Perceptions of Engaging and Maintaining Women in an Intervention

(23) What types of priorities would you want to see addressed through a collective of women? [probe: police, violence, motherhood challenges, debt, social support, issues with relationships with male partners or clients, housing challenges, etc.]

(24) Is there a group of women in the sex trade who you know that you trust and who support or help each other?

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
