# Peer review of "Women Trading Sex in a U.S.-Mexico Border City: A Qualitative Study of the Barriers and Facilitators to Finding Community and Voice"

_sexes, doi:10.3390/sexes1010001_

Round 1
Reviewer 1 Report
Thank you for the opportunity to review your manuscript. It is really interesting and would be a valuable contribution to the field. However, there are ways in which this manuscript can and should be strengthened. Please see the attached document for my recommended revisions.

Reviewer 2 Report
Thank you for the kind invitation to review this manuscript from Mexico regarding Violence and Sexual Exploitation among Women Trading Sex. My comments are provided below for the authors to consider.
Introduction
1.“explore the facilitators and barriers to community mobilization of women trading sex in Tijuana, Mexico in order to inform structural interventions for this vulnerable population, as well as ways to better involve the women in the combat against abuse” - I am not clear about how violence is explored in this study.
2.“little data exist on the potential for mobilization to occur among women engaging in survival sex in the context of substance use and violence in Mexico”. —Please add reference for these little data on abused women.
Adding some research hypotheses would help the readers better understand what the authors seek from the study.
Methods
3. The population under investigation is not just sex workers in Mexico, as indicated in the title. It is a very specific and strictly defined population of sex workers, with certain characteristics (as defined by the project, they were drawn from). To my view, this needs to be clear also in the title and elsewhere.
4. “During the qualitative interviews, the researcher used open‐ended questions, e.g. to understand the participants’ sense of unity with others via collective identity, collective efficacy, and collective agency”. - I suggest the authors to provide the axes of the interview schedule they used during the interview to guide the discussion. It would be also helpful to cite the research they based this interview schedule on.
5. selection criteria for the current study seem to involve many prerequisites regarding drug use. The title is somehow misleading since it refers to violence and sexual exploitation while the study has been based on a drug intervention program. This drug use issue, needs to be somehow taken into consideration from the beginning and incorporated.
6. Please add a short description on the practical aspects of recruitment. How people were approached, by whom and what else did the process include?
7. I haven’t seen any criteria regarding violence and sexual exploitation in the methods section. In the introduction the authors mention that “little data exist on the potential for mobilization to occur among women engaging in survival sex in the context of substance use and violence in Mexico”. The title also places emphasis on that. However, I wonder how the rest of the manuscript relates to these issues.
8. independent street‐based workers represented (58%) at the time of the interview. I wonder what is the percentage in the general population of sex workers in Mexico and how this proportion was this handled during recruitment.
9. As above, those trading sex independently represent 90% of the respondents. I wonder what is the percentage in the general population of sex workers in Mexico and why this parameter was not taken into account during the recruitment.
10. Please add references for “the domains and codes used in this analysis”. Where did the authors base their analytical framework?
Discussion / Conclusion
11. “..our intent was to include and further understand the potential to mobilize substance‐using females especially, a group commonly excluded from networks of women in the sex trade who have organized around their human rights”. - The objectives and the population of interest seem to be different throughout the manuscript. I would suggest the authors to revisit both the study objectives and the population of interest and define them more clearly.
Reviewer 3 Report
I was happy to read this manuscript, as it describes the daily living situations of real people who are experiencing a time sensitive crisis and deserve better.
I was very pleased that the authors worked hard to seek explanatory variables that never blamed the victim (although I recognize that not all sex workers should be considered victims, of course) and, instead, offered interested parties several different entry points for effecting change in the quality of life for these women. And doing so in a way that is not patronizing.
There were miniscule language disfluencies in the manuscript (e.g., the dropping of an indefinite article) that did not detract from the work but would be worth adjusting for the readers sake.
Overall, I am pleased with the quality of the work and they way that it is presented.
Round 2
Reviewer 2 Report
Many thanks for the revised version of the manuscript. Some more comments are provided below for the authors to consider.
Authorship
1. Any additions in the authorship, need to be justified and accepted by the journal’s review committee.
Introduction
2. Although mobilization of women in the sex trade in Mexico occurs, the extent and nature of mobilization in the context of substance use and violence among the most vulnerable women in Tijuana is not well understood. / What does this mean “In the context of substance use and violence..”? How does this translate into the study objectives?
3. Justification of the study is largely based on the lack of evidence regarding substance use and violence (e.g. However, little data exist on the potential for mobilization to occur among women engaging in survival sex in the context of substance use and violence in Mexico.”). However, violence is just presented as a study finding (one of the barriers identified) and not as a crucial characteristic guiding the sampling design of being the focus of the study. Please check the whole manuscript and treat the aspect of violence as a finding and not as a study objective or as a characteristic of the sample.
Methods
4. Methods have been improved and seem to be very inclusive now. However, it seems from the “conclusion section” that the study was only representative of substance-using individuals. This is what the authors mention in text. I don’t see this aspect very clear in the sampling design. (For example, I don't see substance use as an inclusion criterion in the following text……… “To be eligible for the parent study, the participants had to be at least 18 years old, exchanged sex for money, goods, or shelter four or more times with four or more clients in the previous 30 days; willing to undergo behavioral surveys (CAPI) and HIV/STI test in the parent grant; agree to receive STI treatment if positive from medical test, and lived in Tijuana, Mexico with no plans to leave the area within 18 months of the first interview date. At least 20% reported substance use (excluding marijuana) and 20% injecting drug use in the past 30 days”//. Please explain or revise.
General
5. Overall, I see that the study has a very specific sample of individuals and explores their barriers and facilitators for mobilization. This is very clear at the results/discussion but at some points, especially at the introduction, there is deviation from the main objectives. I would suggest the authors to revisit the introduction and make sure that it appropriately addresses the gap in literature and the need to increase understanding regarding “barriers to mobilization of substance using individuals that trade sex?”.
